# Fertility and Reproduction after Recovery from Anorexia Nervosa: A Systematic Review and Meta-Analysis of Long-Term Follow-Up Studies

**DOI:** 10.3390/diseases8040046

**Published:** 2020-12-16

**Authors:** Rayane Chaer, Nour Nakouzi, Leila Itani, Hana Tannir, Dima Kreidieh, Dana El Masri, Marwan El Ghoch

**Affiliations:** Department of Nutrition and Dietetics, Faculty of Health Sciences, Beirut Arab University, P.O. Box 11-5020 Riad El Solh, Beirut 11072809, Lebanon; Rayane.Chaer@outlook.com (R.C.); nour_nakouzi99@hotmail.com (N.N.); l.itani@bau.edu.lb (L.I.); hana.tannir@bau.edu.lb (H.T.); d.kraydeyeh@bau.edu.lb (D.K.); dana.masri@bau.edu.lb (D.E.M.)

**Keywords:** anorexia nervosa, eating disorders, fertility, menses, reproductive function, weight restoration

## Abstract

Reproductive health is compromised during anorexia nervosa (AN). However, it is still unclear whether this medical complication is reversible after recovery from AN. The purpose of this paper was to conduct a systematic review of the major reproductive health outcomes in females after recovery from AN. The review was conducted in adherence to preferred reporting items for systematic review and meta-analyses (PRISMA) guidelines. Data were collated using meta-analysis and a narrative approach. Of the 1186 articles retrieved, five studies met the inclusion criteria and were reviewed. These studies monitored weight-restored females who had recovered from AN for a follow-up period of between six and 18 years. Their narrative analysis revealed that appropriate treatment of AN leads to the normalization of reproductive function, especially in terms of fertility, pregnancy, and childbirth rates. The meta-analysis confirmed this finding, where the pooled odds of childbirth rates between the AN group and the general population was not statistically significant (OR = 0.75, 95% CI: 0.43–1.29, *p* = 0.41). We conclude that if patients undergo appropriate eating-disorder treatment and weight restoration, it appears to be unlikely that reproductive health is affected by AN. However, since this finding is derived from only a few studies, it requires replication and confirmation.

## 1. Introduction

Anorexia nervosa (AN) is a common health problem during adolescence and young adulthood with a lifetime prevalence of roughly 0.6% [1,2] known to be associated with considerable physical health [3,4], psychosocial comorbidities [5,6], and an increase in the mortality rate [7]. According to the fifth edition of the Diagnostic and Statistical Manual of Mental Disorders (DSM-5), to be diagnosed with AN, the following criteria must be met [1]: (i) restriction of energy intake relative to requirements leading to significantly low bodyweight in the context of age, sex, developmental trajectory, and physical health; (ii) intense fear of gaining weight or becoming fat, even though underweight; and (iii) Disturbance in the way in which one’s bodyweight or shape is experienced, undue influence of bodyweight or shape on self-evaluation, or denial of the seriousness of the current low bodyweight [1]. A considerable amount of data has been derived from research into how reproductive health is negatively affected during AN [8,9]. This seems to occur through two mechanisms that often operate in concert: under-eating and low bodyweight [10], which unavoidably lead to secondary amenorrhea [11], as well as infertility defined as 1 year of unwanted non-conception with unprotected intercourse in the fertile phase of the menstrual cycles [12], and therefore, reproductive function is compromised [13]. These findings have also been found in AN-animal models that were induced through food restriction that determined significant weight loss and led to the blockage of the reproductive function [14], but when these mice were put on recovery protocol, with adequate food intake and normalization of bodyweight, this determined the restoration of the normal reproductive function [14], however, this latter finding has not been extended to humans. Recent large-sample studies examined long-term consequences of AN on reproductive health outcomes [15,16], comparing individuals with a history of AN vs. their counterpart controls without eating disorders. These studies found that reproductive health is mainly compromised in females with a history of AN [15,16]. The discrepancy between human and animal models seems to be due to the fact it was unclear whether these patients [15,16] had recovered from AN or were still affected. It is more likely that these samples were heterogeneous and included recovered and non-recovered patients [15,16]. In fact, little is known about what happens after recovery from AN, prompting the legitimate clinical question:

### Is the Impairment of Reproductive Function during AN Reversible after Recovery?

This issue is of clinical relevance for the health professionals who usually treat eating disorders (i.e., physicians, psychologists, dieticians, etc.) as well as for patients affected by AN. To the best of our knowledge, no systematic review addressing this issue as a primary outcome has yet been conducted in order to provide a less biased interpretation of the evidence published to date. In light of these considerations, we set out to systematically review published literature on this topic in accordance with the PICO process [17], as detailed below:

P—Population: female adolescents [18] or young adults who had a reliable diagnosis of AN (i.e., based on the Diagnostic and Statistical Manual of Mental Disorders (DSM), International Classification of Diseases (ICD), etc.) [19]; I—Intervention: any treatment for AN, that determined recovery; C—Comparison: AN group and healthy control group (when available); O—Outcome: major outcomes related to reproductive health, in particular fertility, pregnancy, and childbirth rates.

## 2. Materials and Methods

The current paper has been completed according to the preferred reporting items for systematic review and meta-analyses (PRISMA) guidelines [20] (Appendix A).

### 2.1. Inclusion and Exclusion Criteria

All studies evaluating any reproductive function outcome in adolescent [21] and young adult females of fertile age with AN were included, provided that they met the following criteria: (i) they were written in English, (ii) they were original articles on studies with a longitudinal design, and (iii) they related to prospective or retrospective observational (analytical or descriptive), experimental or quasi-experimental controlled or non-controlled studies, documenting significant weight gain or weight restoration in patients. No reviews, cross-sectional studies, or non-original articles (i.e., case reports, editorials, ‘Letters to the Editor’ or book chapters) were included.

### 2.2. Information Source and Search Strategy

The literature search was designed and performed independently in duplicate by three authors, namely, the principal investigators (RC and NN) and the senior investigator (ME). The PubMed/Medline [22], the Cochran Library [23], and ScienceDirect [24] databases were systematically screened using MeSH terms with combinations as follows:

#1 = fertility, #2 = infertility, #3 = pregnancy, #4 = pregnancy outcomes, #5 = parity, #6 = fecundity, #7 = reproduction, #8 = number of children, #9 = bulimia, #10 = bulimia nervosa, #11 = restored weight of anorexia, #12 = restored weight anorexia nervosa, #13 = restored weight of anorexics, #14 = regained weight of anorexia, #15 = regained weight of anorexia nervosa, #16 = regained weight of anorexics, #17 = recovered weight of anorexia, #18 = recovered weight of anorexia nervosa, #19 = recovered weight of anorexics, #20 = history of anorexia, #21 = history of anorexia nervosa, #22 = history of anorexics, #23 = following anorexia, and #24 = following anorexia nervosa, together with the combinations ((#1 OR #2 OR #3 OR #4 OR #5 OR #6 OR #7 OR #8) AND (#11 OR #12 OR #13 OR #14 OR #15 OR #16 OR #17 OR #18 OR 319 OR #20 OR #21 OR #22 OR #23 OR #24)) OR ((#1 OR #2) AND (#9 OR #10)).

Moreover, a manual search was carried out to retrieve other articles that had not been identified via the initial search strategy. The publication date was not considered an exclusion criterion for the purposes of this review.

### 2.3. Study Selection

Two authors independently screened the resulting articles for their methodology and appropriateness for inclusion. Non-controlled studies were selected for quality appraisal according to the National Institute for Health and Clinical Excellence (NICE) guidelines checklist, in which a total score of 0–3 indicates poor quality, between 4–6, fair quality, and ≥7, good quality [25]. In controlled studies, a quality appraisal was conducted according to the Newcastle-Ottawa Scale (NOS), which relies on a 9-star system whereby scores of 0–3, 4–6, and 7–9 are considered poor, moderate, and good quality, respectively [26]. Consensus discussion was used to resolve disagreements between authors.

### 2.4. Data Collection Process and Data Items

The title and abstract of each paper were firstly assessed by two independent authors for language suitability and subject matter relevance. The studies selected in this way were then assessed in terms of their appropriateness for inclusion and the quality of the method.

Details of the studies that passed both rounds of screening are shown in Table 1, including the name of the first author, year conducted, design of the study, sample size, duration of follow-up, and finding of each study.

### 2.5. Data Synthesis

All the studies that met the inclusion criteria have been presented as a narrative synthesis [27]. Moreover, the effect size of interest was odds of childbirth rate in those with AN compared to controls. A meta-analysis was performed to calculate pooled odds from the studies included in the analysis. Heterogeneity was assessed using I^2^ statistic, and acceptable heterogeneity was determined at I^2^ < 60%. A forest plot was used to illustrate odds of childbirth rates using Review Manager 5 software (RevMan 5.3, Copenhagen D, Cochrane Collaboration: London, UK) [28].

## 3. Results

The initial search across the three databases retrieved 1186 papers. After the first round of screening (titles and abstracts), 503 papers were excluded on the following grounds: not in English, not conducted on humans, or were considered as duplicates. The second round of screening excluded: inappropriate types of papers that were different from the original research article (i.e., review articles, clinical case reports, chapters from books, and ‘Letters to the Editor’) or were not available abstract and/or full text (*n* = 377). Of the remaining 306 articles screened according to methodology and content, 301 papers were excluded on the following grounds: (i) dealing with eating disorders, but not AN, (ii) about patients with AN, but not related to reproductive health outcomes, (iii) conducted on patients with AN on reproductive health outcomes, but with a cross-sectional design or with unavailable follow-up data, (iv) composed of non-homogenous samples, in other words, recovered and weight-restored patients with AN as well as patients who were still actively affected by AN.

Thus, at the end of the screening process, only five articles were available for systematic review that underwent a narrative synthesis (Figure 1). According to the NICE checklist, the non-controlled studies (*n* = 3) were of fair quality (mean score 5.0 points) (Table 2), whilst the NOS checklist indicated that the controlled studies (*n* = 2) were of moderate quality (mean score 4.5 points) (Table 3).

In 1986, Kohmura et al. [29] carried out a non-controlled study that assessed the reproductive function of 21 patients with AN (average age of 20.8 ± 0.7 years) after an average of 10.2 years of follow-up (range eight to 13 years). The bodyweight of 19 patients was restored to within normal range. Seventeen of those had a ROM and recovery from an eating disorder. Of these patients, 16 were married after treatment, 14 of these became pregnant, and 12 delivered a child. The authors concluded that most patients are able to have a child after appropriate treatment for AN [29]. In fact, the present study showed that the rate of pregnancy after treatment (87.5%) was comparable to that in the general population.

Two years later, in 1988, Brinch et al. [30] used a non-controlled design to study the reproductive function of 141 patients (mean age at the primary contact 19.2 years) with a history of AN over a follow-up of 12.5 years (mean age at follow up 32.7 years). Fifty females became pregnant and gave birth to a total of 86 children, and an average of 1.7 birth, similar to the childbirth rate in Denmark in that year and with mean ages at first delivery was 26.1 years compared to 24.1 years in the general Danish population.

Only 10% of the patients interviewed had problems with fertility. This was no different from the infertility rates in the general population. Despite this fact, Brinch et al. found that the perinatal mortality in their sample was six times higher than in the general population and low birth weight was more than twice as likely [30]. Thirty-six mothers out of 50 completely recovered from AN and conceived spontaneously without any hormonal treatment.

In 1994, Shomento and Kreipe [31] carried out a non-controlled retrospective study that aimed to determine the bodyweight (expressed as % normal bodyweight) at which ROM occurred after recovery from AN and the subsequent effects on fertility. The sample was composed of 83 adolescent females with a diagnosis of AN based on the DSM-III-R criteria for the treatment of AN, who were interviewed at least four years after hospitalization with an average age at follow up of 22.2 ± 3.7 years. The authors found that following AN, ROM can be expected when bodyweight reaches 92% of normal weight for height (*n* = 54). Moreover, all females (*n* = 14) who wanted to conceive were able to do so within one year of trying to become pregnant. The authors concluded that a history of AN is unlikely to affect fertility if weight is restored to normal [31].

Five years later, in 1999, Bulik et al. [32] used a controlled study to retrospectively examine the fertility and reproductive function in 66 females with a mean age at interview of 32.4 ± 8.0 years and a history of AN based on DSM-III-R. The controls were 98 randomly selected healthy females from the community with a mean age at interview of 35.5 ± 6.2 years. Of the females with a history of AN, 45 out of 66 delivered, compared to 76 out of 98 females in the control group. Females in both groups did not differ in the rate of pregnancy, the mean number of pregnancies per woman (1.8 ± 2.3 vs. 1.9 ± 1.4; *p* < 0.05), or age at first pregnancy (24.5 ± 4.7 vs. 23.7 ± 4.7 years; *p* > 0.05). However, females with AN had significantly more miscarriages (17 vs. 12; *p* = 0.006) and more Caesarean deliveries (7 vs. 2; *p* = 0.01). This study could have been excluded on the grounds that the sample included patients with active AN. However, the authors clearly stated that 75–80% of their sample at the moment of evaluation was free from AN. Due to the paucity of studies on the topic, we decided to include it.

In 2009, Wentz et al. [33] used a controlled design to study the reproduction function in 48 females with AN interviewed 18 years after the onset of AN at a mean age of 32.4 years. They were matched with a community comparison group. Six out of 48 females from the AN group still had an eating disorder, and none of these had become a mother. Comparing the pregnancy rates of 27 out of 42 females from the AN group and 31 out of 48 females from the control group, there was no significant difference between the two groups in childbirth rates. The authors conclude that adults who had recovered from teenage-onset AN did not differ in most aspects from matched controls with respect to pregnancies and the development of their offspring [33].

### Meta-Analysis

Figure 2 shows the effect sizes and 95% CI of the two studies included in the meta-analysis. The pooled odds ratio of childbirth rates between the two groups (AN group vs. the general population) was not statistically significant (OR = 0.75, 95% CI: 0.43–1.29, *p* = 0.41), and a heterogeneity analysis revealed I^2^ = 0%. 

## 4. Discussion

The aim of the current systematic review was to provide evidence on the reproductive health outcomes after treatment in female patients with AN. Five studies, comprising 353 females were reviewed. The majority of women in these studies had recovered from AN and were followed-up over the long term. These studies were objectively judged to be of fair-moderate quality.

Despite the paucity of studies, a common finding unifies the included studies; the normalization of reproductive function seems to be possible and after weight gain/restoration and recovery from AN. Our findings disagree with the results derived from two large-sample controlled studies [15,16] that reported a permanent impairment of reproductive function in females with a history of AN, which in turn resulted in reductions and delays in reproduction [15,16]. However, it is difficult to determine whether patients with a history of AN included in these two studies continued to have an eating disorder at the time of the infertility evaluation [15,16]. This is likely to be the case, since the inclusion of patients with active AN together with those who had already recovered may have inflated the rates of reproductive dysfunction and infertility in this population. Finally, confirmation of our findings can be found in earlier studies. Despite not considering reproductive outcomes as a primary factor, these earlier studies reported that the rates of pregnancy in their samples after treatment of AN were between 80% and 90% [34,35,36].

### 4.1. Clinical Implications

Our findings have two main clinical implications. Firstly, clinicians should be aware of the importance of bodyweight restoration; if treated, AN does not preclude the possibility of subsequent normalization of reproductive health outcomes. Secondly, this finding should be openly discussed with patients with AN, especially those who are starting an eating disorder treatment.

Indeed, this accords with the recommendations of experts who underline the importance of nutritional intervention and weight restoration as the cornerstone strategy for normalization of medical [4,37,38] and psychological [5] alterations related to being underweight, suffering from malnutrition, and under-eating.

### 4.2. Strengths and Limitations

To the best of our knowledge, this systematic review is the first to assess the status of reproductive health in patients after recovering from AN, over a relatively long-term follow-up period (six to 18 years). However, due to certain limitations, our findings should be interpreted with caution. In particular, only a few studies have been included. In addition, the small sample size was a common feature of most of the studies reviewed. Moreover, it is unclear whether the patients included in our systematic review were supposed to have recovered from AN and who were monitored for reproductive outcomes, had achieved ROM or not. Indeed this is of clinical relevance since a subgroup of women who have recovered from AN and achieved a normalization of bodyweight would still be experiencing amenorrhea and may have required treatment with gonadotropins, or/and in vitro fertilization to obtain certain benefits [37]. Finally, not all included studies (only one) clearly mentioned the bodyweight status of the newborn, since a low birth-weight is known to be associated with long-term morbidity and health consequences (i.e., metabolic disorders) of the offspring at adult age [39].

## 5. Conclusions and Areas for Future Research

If female patients undergo appropriate eating disorder treatment and weight restoration, it appears unlikely that fertility is affected by AN. The recovered female patients do not seem to differ in most aspects from matched healthy controls with respect to pregnancies and the development of their offspring. However, since this conclusion is derived from a limited number of studies, further research is necessary to replicate this finding, as well as taking into account other long-term outcomes (i.e., reproduction lifespan, health status of offspring, etc.). Moreover, it is vital that more research is carried out on fertility in males with AN to better understand the impact of the latter on reproductive health in this population, which seems to be more complex than females and still not fully understood.

## Figures and Tables

**Figure 1 diseases-08-00046-f001:**
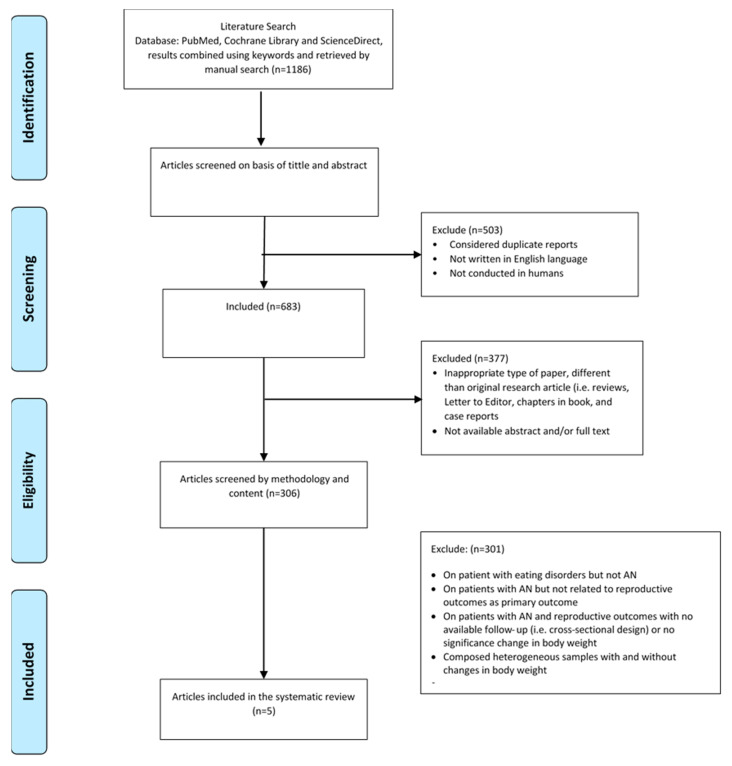
The flowchart summarizing the study selection procedure.

**Figure 2 diseases-08-00046-f002:**
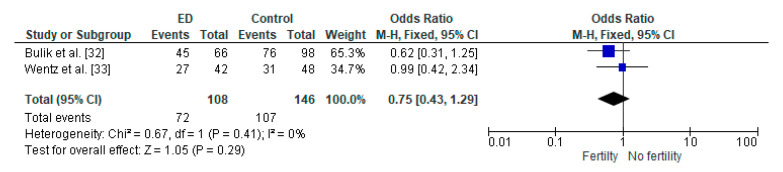
Forest plot of odds of childbirth rates in females with AN compared to controls.

**Table 1 diseases-08-00046-t001:** Studies included in the systematic review.

First Author	Year	Study Design	Sample	Follow-Up	Finding
Kohmura et al. [29]	1986	Non-controlled study	N = 21	8–13 years (Average 10.2 years)	17 had ROM, and 16 were married after treatment 14 of these became pregnant and 12 delivered
Brinch et al. [30]	1988	Non-controlled study	N = 141	4–22 years (Average 12.5 years)	50 pregnant and gave birth to 86 children Similar infertility rate to the general population in Denmark Similar childbirth rate to the general population in Denmark
Shomento et al. [31]	1994	Non-controlled study	N = 83	6 years	54 had ROM that occurred at 92% of normal weight for height All 14 females willing to conceive were able to do so within 1 year of trying to become pregnant
Bulik et al. [32]	1999	Controlled study	N = 66 (AN patients) N = 98 (Controls)	>10 years	45 in AN group delivered, and 76 the control group delivered, and both groups did not differ on main reproductive outcomes AN group had significantly more miscarriages (17 vs. 12; *p* = 0.006) and Caesarean deliveries (7 vs. 2; *p* = 0.01)
Wentz et al. [33]	2009	Controlled study	N = 42 (AN patients) N = 48 (Controls)	18 years	27 in the AN group and 31 in the control group had children Adults recovered from AN did not differ in most aspects from matched controls with respect to pregnancies and development of offspring

ROM = resumption of menses; AN = anorexia nervosa.

**Table 2 diseases-08-00046-t002:** NICE guidelines quality assessment checklist.

Author	Kohmura et al. (1986) [29]	Brinch et al. (1988) [30]	Shomento et al. (1994) [31]
Case series collected in more than one center, i.e., multi-center study	0	0	0
Is the hypothesis/aim/objective of the study clearly described?	1	1	1
Are the inclusion and exclusion criteria (case definition) clearly reported?	0	0	0
Is there a clear definition of the outcomes reported?	1	1	1
Were data collected prospectively?	1	1	0
Is there an explicit statement that patients were recruited consecutively?	0	0	1
Are the main findings of the study clearly described?	1	1	1
Are outcomes stratified?	1	1	1
Total Score	5	5	5

Yes = 1; No = 0; Total score 0–3 = poor quality; 4–6 = moderate quality; 7–9 = good quality.

**Table 3 diseases-08-00046-t003:** Newcastle-Ottawa Scale (NOS) quality assessment checklist.

Author	Bulik et al. (1999) [32]	Wentz et al. (2009) [33]
Selection		
Represents cases with independent validation	1	0
Cases are consecutive	0	0
Controls are from the community	1	1
Controls have no history of anorexia nervosa	1	1
Comparability		
Controls are comparable for the most important factors	1	1
Control for any additional factor	0	0
Ascertainment of exposure		
Same method of ascertainment for cases and controls	1	1
Cases and controls have completed follow up	0	0
Total Score	5	4

Yes = 1; No = 0; Total score 0–3 = poor quality; 4–6 = fair quality; 7–9 = good quality.

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
