# Peer review of "Fertility and Reproduction after Recovery from Anorexia Nervosa: A Systematic Review and Meta-Analysis of Long-Term Follow-Up Studies"

_diseases, 2020, doi:10.3390/diseases8040046_

Round 1
Reviewer 1 Report
The paper is very interesting. The authors perform a big work to select the literature's data
Author Response
The paper is very interesting. The authors perform a big work to select the literature's data.
Response: We thank the reviewer for the appreciation.
Reviewer 2 Report
Chaer et al., presented a nice review of how anorexia could impact fertility after recovery. As the authors mentioned there is a big gap in knowledge and many of the published papers lack of reproducibility, they have very small sample size, etc. etc. These problems made that after selection the authors ended with only 5 papers. I have a few comments that could improve the paper. 1. Does age of the patients is a critical factor for the pregnancy outcome? Please include the ages of the patients. DO you have information about menopause in these patients? Are these patients going to IVF cycles. 2. The main purpose of the paper is to talk about the what is happening in human, but it will be great if you add a section where you analyze and discuss from animal models. 3. in the introduction please define properly what is anorexia according to DMS5, please also include the definitions of infertility, subfertility etc. 4. Can you include a small section about fertility in males? that is even more complicated but it will be great if you discuss some of the data on your discussion section 5. Are the lifespans and reproductive lifespan equal in controls vs recovered patients?Author Response
Reviewer 2#
Chaer et al. presented a nice review of how anorexia could impact fertility after recovery. As the authors mentioned there is a big gap in knowledge and many of the published papers lack of reproducibility, they have very small sample size, etc. etc. These problems made that after selection the authors ended with only 5 papers. I have a few comments that could improve the paper.
- Does age of the patients is a critical factor for the pregnancy outcome? Please include the ages of the patients. Do you have information about menopause in these patients? Are these patients going to IVF cycles.
Response: The age of participants has been included in the text wherever available (Lines 152, 160, 176, 181, 183 and 192). No information about the menopause of these patients neither if underwent IVF cycles. However we added this in the Discussion section as limitations (Lines 239 – 244) and new direction for future research (Lines 253 – 254).
- The main purpose of the paper is to talk about what is happening in human, but it will be great if you add a section where you analyse and discuss from animal models.
Response: We thank the reviewer for the valuable comment. Now we added a section on animal model in the Introduction section (Lines 47 – 53 and 55 – 56).
- In the introduction please define properly what is anorexia according to DMS5, please also include the definitions of infertility, subfertility etc.
Response: Now in the Introduction section we added the definition of anorexia nervosa according to DSM 5 as well as infertility and Done as suggested (Lines 35 – 42 and 45 – 46)
- Can you include a small section about fertility in males? That is even more complicated but it will be great if you discuss some of the data on your discussion section.
Response: We added as suggested few statements in the Discussion section (Line 254 – 257).
- Are the lifespans and reproductive lifespan equal in controls vs. recovered patients?
Response: The lifespans and reproductive lifespan are not available in the included studies. However we added this in the Discussion section (Line 254).
Reviewer 3 Report
The Authors did not report the PRISMA checklist. They did not report any description of meta-analytic methods and Figure 2 was not included in the manuscript.
Author Response
The Authors did not report the PRISMA checklist.
Response: The PRISMA checklist has been added as Supplementary file (Suppl. 1).
They did not report any description of meta analytic methods and Figure 2 was not included in the manuscript.
Response: A description of the meta-analysis has been reported (Lines 122 – 126) and included Figure 2 (Page 7).
Reviewer 4 Report
Fertility and Reproduction after Recovery from Anorexia Nervosa: A Systematic Review and Meta4 analysis of Long-term Follow-up Studies
Abstract – good
Intro – good
Materials/Methods – I suggest putting line 102 below the table, not above it.
Table 1 – first row – Right column: States “17 with” had ROM. Please specify what “17 with” means. All patients had AN, so “with what”?
Third row – Shomento study – Please state how many women conceived within that 1 year period. (body states that this was 14 patients)
Results – line 125 – “according to the NICE checklist”. Please give the full name referred to by the acronym, and specify that this is the NICE quality appraisal checklist
Line 145 – Brinch study – please state whether Brinch examined whether these patients had a higher use of ART in conceiving. Also, state whether weight gain during pregnancy was or was not reported in that study.
Line 167 – suggest deleting “remaining”, and inserting “pregnancy rates of”
Discussion –
2 comments:
First – please state whether each of these patients who were monitored for reproductive outcomes in the studies had ROM. This is very important, of course, because women with AN who have achieved a normal BMI often still do not menstruate, and will require treatment with gonadotropins, +/- IVF. This would be appropriately included in ‘strengths and limitations’.
Second – please determine if weight gain during pregnancy was included in any of the studies and report it. This is of great interest, because anecdotally it appears that often these women do NOT achieve adequate weight gain. Poor weight gain in pregnancy leads to epigenetic changes which predispose to Metabolic syndrome and other adverse health conditions.
Author Response
Abstract – good
Response: We thank the reviewer.
Intro – good
Response: We thank the reviewer.
Materials/Methods – I suggest putting line 102 below the table, not above it.
Response: Now we put the line 102 below the table.
Table 1 – first row – Right column: States “17 with” had ROM. Please specify what “17 with” means. All patients had AN, so “with what”?
Response: We corrected the typos mistake. We meant 17 had ROM (Table 1).
Third row – Shomento study – Please state how many women conceived within that 1-year period. (body states that this was 14 patients)
Response: Now we stated the number of women conceived within 1-year period (Table 1).
Results – line 125 – “according to the NICE checklist”. Please give the full name referred to by the acronym, and specify that this is the NICE quality appraisal checklist
Response: Now we added the full name of the abbreviation that stands for NICE (Line 105).
Line 145 – Brinch study – please state whether Brinch examined whether these patients had a higher use of ART in conceiving. Also, state whether weight gain during pregnancy was or was not reported in that study.
Response: The 36 recovered women from AN, conceive spontaneously without any hormonal treatment and this information has been mentioned in the text (Line 169 – 170). The weight gain during pregnancy was or was not reported.
Line 167 – suggest deleting “remaining”, and inserting “pregnancy rates of”
Response: done as suggested (Line 194).
Discussion – 2 comments:
First – please state whether each of these patients who were monitored for reproductive outcomes in the studies had ROM. This is very important, of course, because women with AN who have achieved a normal BMI often still do not menstruate, and will require treatment with gonadotropins, +/- IVF. This would be appropriately included in ‘strengths and limitations’.
Response: We thank the reviewer for the valuable comment that we included in the Strengths and limitations subsection (Lines 239 – 244).
Second – please determine if weight gain during pregnancy was included in any of the studies and report it. This is of great interest, because anecdotally it appears that often these women do NOT achieve adequate weight gain. Poor weight gain in pregnancy leads to epigenetic changes which predispose to Metabolic syndrome and other adverse health conditions.
Response: We thank the reviewer for the valuable comment that we included in the Strengths and limitations subsection (Lines 244 – 247).
Round 2
Reviewer 2 Report
The authors answered and corrected all my previous comments
Reviewer 3 Report
None